# Functional traits and phenotypic plasticity modulate species coexistence across contrasting climatic conditions

Ignacio M. Pérez-Ramos[1], Luis Matías [2,3], Lorena Gómez-Aparicio[1] & Óscar Godoy [4]

Functional traits are expected to modulate plant competitive dynamics. However, how traits and their plasticity in response to contrasting environments connect with the mechanisms determining species coexistence remains poorly understood. Here, we couple field experiments under two contrasting climatic conditions to a plant population model describing competitive dynamics between 10 annual plant species in order to evaluate how 19 functional traits, covering physiological, morphological and reproductive characteristics, are associated with species' niche and fitness differences. We find a rich diversity of univariate and multi-dimensional associations, which highlight the primary role of traits related to water- and light-use-efficiency for modulating the determinants of competitive outcomes. Importantly, such traits and their plasticity promote species coexistence across climatic conditions by enhancing stabilizing niche differences and by generating competitive trade-offs between species. Our study represents a significant advance showing how leading dimensions of plant function connect to the mechanisms determining the maintenance of biodiversity.

[1] IRNAS, CSIC, LINC Global, Av. Reina Mercedes 10, 41012 Sevilla, Spain. [2] Departamento de Biología Animal, Biología Vegetal y Ecología, Universidad de Jaén, Ed. B3, Paraje las Lagunillas SN, E-23071 Jaén, Spain. [3] Departamento de Biología Vegetal y Ecología, Universidad de Sevilla, Av. Reina Mercedes, Sevilla E-41080, Spain. [4] Departamento de Biología, Instituto Universitario de Investigación Marina (INMAR), Universidad de Cádiz, E-11510 Puerto Real, Spain. Correspondence and requests for materials should be addressed to Ó.G. (email: oscar.godoy@uca.es)

Ecologists seek to understand the mechanisms that govern the assembly of plant communities[1]. Achieving this goal is often a daunting task given the difficulty to estimate the plethora of interactions among co-occurring species, as well as the potential influence that changes in environmental conditions might have on these interactions[2,3]. A generally appreciated alternative to tackle this complex network of plant–plant and plant–environment interactions uses species' traits rather than relying on their taxonomical identity[4,5]. There is growing evidence of the relevance of functional traits for determining the three main components of individual performance—growth, reproduction, and survival[6–8], as well as the strength and sign of plant interactions[9–11]. However, it is still poorly understood how interspecific trait differences connect with the stabilizing and equalizing mechanisms that jointly determine biodiversity maintenance[12,13]. Establishing such connections is critical to support a theoretically justified trait-based approach that can be applied to different research areas of community ecology[5,14].

Our poor understanding on this topic comes from the fact that prior work has found several correlations between single interspecific trait differences and the competitive asymmetries that drive species exclusion, but none associated with the niche differences that stabilize species coexistence. For instance, higher wood density has been correlated with greater competitive tolerance in trees[11], and deeper root systems have been related to higher species' fitness in annual plants[14]. These are counterintuitive results because differences in both traits were traditionally thought to play a major role as drivers of resource partitioning and, therefore, of niche differentiation across different environmental conditions (e.g. wood density[15]) and soil layers (e.g. rooting depth[16]). So far, niche differences have been only detected when considering multiple axes of trait variation together (i.e. phenology, seed size, height, specific root length), suggesting that the maintenance of species diversity has a multidimensional nature[14].

In addition, we suffer from the limitation that interspecific trait differences have been related to the determinants of competitive outcomes under single environmental conditions[14]. This is indeed an unrealistic scenario because plant competitive dynamics commonly occur under contrasting conditions. Species respond to environmental changes by means of multiple morphological and physiological trait adjustments that serve to alleviate stress levels and to increase the uptake of limiting resources[17,18]. Yet, it is unknown how the species' ability to vary their phenotypic expression across environments (i.e. phenotypic plasticity) modulates the outcome of plant competitive interactions and their likelihood to coexist[19].

These two main knowledge gaps highlight the necessity of carrying out studies that consider a larger array of functional traits and environmental conditions in order to capture a broader range of ecological dimensions that allow us to detect and to distinguish the univariate and multidimensional influence of functional traits as drivers of stabilizing niche differences. Specifically, previous studies have been mainly focused on morphological (mostly aboveground) traits because they are easily measurable or they are readily available in worldwide databases, leaving out the critical importance of physiological traits in mediating species coexistence[20].

Here, we present the results from a field-competition experiment with 10 annual plant species typical of Mediterranean grasslands in which we measured 19 functional traits related to plant physiology, morphology, phenology, and reproductive ability (Table 1) under two contrasting scenarios of water availability. We did so by sowing species seeds right after autumn rains (control treatment) and 64 days after the start of the rainy season, which caused a precipitation reduction of 206 mm

(drought treatment see the section "Methods" for more details). The aim of the study is two-fold. First, we aim to evaluate whether single functional traits can capture the demographic signature of stabilizing niche differences. We also expect that single functional traits are associated with the differences in competitive ability that hinder coexistence between species. Particularly, we are interested here in documenting trade-offs across climatic conditions, following the idea that a trait value favorable in one environment might prove unfavorable in another one[21]. For instance, it is likely that species exhibiting traits associated with a more efficient use of resources (i.e. drought-tolerant species) show higher competitive superiority under more stressful conditions but not necessarily under more favorable environments. Second, we further aim to assess whether phenotypic plasticity in response to drought promotes or hinders coexistence by increasing or decreasing interspecific differences in niche and competitive ability between species. Prior work in the literature clearly establishes the hypothesis that more plastic species will likely show similar fitness across a broader range of conditions, which might allow to maintain the dominance of the most competitive species (Jack-of-all-trades strategy[22]). However, there is no prior information to expect a particular relationship between trait plasticity and niche variation across contrasted environments[19]. The effort of combining theoretical advances in modern coexistence theory with experimental evaluation of species competition using multiple traits under contrasting climatic conditions aims to increase our understanding on the functional mechanisms of community assembly under climate-induced shifts that might occur in the near future. In fact, global change models predict more severe and recurrent drought periods for the next decades in many temperate and Mediterranean ecosystems[23]. But equally important, this study represents a rigorous test to analyze whether trait-based approaches can be used as valuable tools to predict the nature and strength of competitive interactions under current and future climatic scenarios. Here, we report evidence of the importance of considering physiological traits related to water-use and light-use efficiency for better understanding community assembly dynamics. This is because variation in such traits enhances stabilizing niche differences and generates competitive trade-offs between species. These results provide a direct pathway linking species' physiology to the mechanisms determining diversity maintenance.

## Results and discussion

**Functional drivers of niche and fitness differences across environments.** As detailed in the section "Methods", we field-parameterized mathematical models of competition with species vital rates and interaction coefficients that allow estimating the processes of stabilizing species coexistence by causing intraspecific competition to be greater than interspecific competition (i.e. niche differences, $1-\rho$), and those that lead to competitive superiority and hence species exclusion in the absence of niche differences (i.e. average fitness differences, $\frac{\kappa_j}{\kappa_i}$)[24]. This modeling approach served to identify single and multiple correlates of interspecific trait differences and their plasticity with the determinants of competitive outcomes, and showed simultaneously the importance for plant competition of different suites of traits across climatic treatments (Fig. 1).

Differences in several individual physiological traits involved in light and soil resource acquisition (i.e. maximum photosynthetic rate—$A_{max}$, stomatal conductance—$g_s$, and leaf N) significantly correlated with average fitness differences between species, yet the sign of such correlations changed under the two climatic treatments (Fig. 1a). Under control rainfall conditions, competitive superiority (that is, having higher fitness than a competitor)

| | Functional traits | Abbreviation | Unit | Leading functional dimension |
|---|---|---|---|---|
| **Table 1 Functional traits quantified in this study, with their abbreviations and units when relevant** | | | | |
| Whole-plant traits | Plant height | – | cm | Light interception |
| | Plant volume | – | $cm^3$ | Light interception |
| Morphological traits | Leaf size | – | $cm^2$ | Plant economics |
| | Specific leaf area | SLA | $cm^2\,g^{-1}$ | Plant economics |
| | Leaf dry matter content | LDMC | $mg\,g^{-1}$ | Plant economics |
| | Root diameter | – | mm | Plant economics |
| | Specific root area | SRA | $cm^2\,g^{-1}$ | Plant economics |
| | Root density | – | $g\,cm^{-3}$ | Plant economics |
| Physiological leaf traits | Max. photosynthesis | $A_{max}$ | $mmoles\,m^{-2}\,s^{-1}$ | Plant growth |
| | Stomatal conductance | $g_s$ | $mmoles\,m^{-2}\,s^{-1}$ | Plant growth |
| | Convexity | – | – | Plant growth |
| | Light compensation point | – | $mmoles\,m^{-2}\,s^{-1}$ | Plant growth |
| | Light saturation point | – | $mmoles\,m^{-2}\,s^{-1}$ | Plant growth |
| | Leaf nitrogen content | LNC | $mg\,g^{-1}$ | Plant economics |
| | Leaf carbon content | LCC | $mg\,g^{-1}$ | Plant economics |
| | Carbon isotope ratio | $\delta^{13}C$ | ‰ | Plant economics |
| | Nitrogen isotope ratio | $\delta^{15}N$ | ‰ | Plant economics |
| Reproductive traits | Reproductive phenology | – | Days | Reproductive ability |
| | Seed mass | – | g | Reproductive ability |

The main functional dimension (light interception, reproductive ability, plant economics spectrum, and plant growth) are indicated for each of the 19 traits

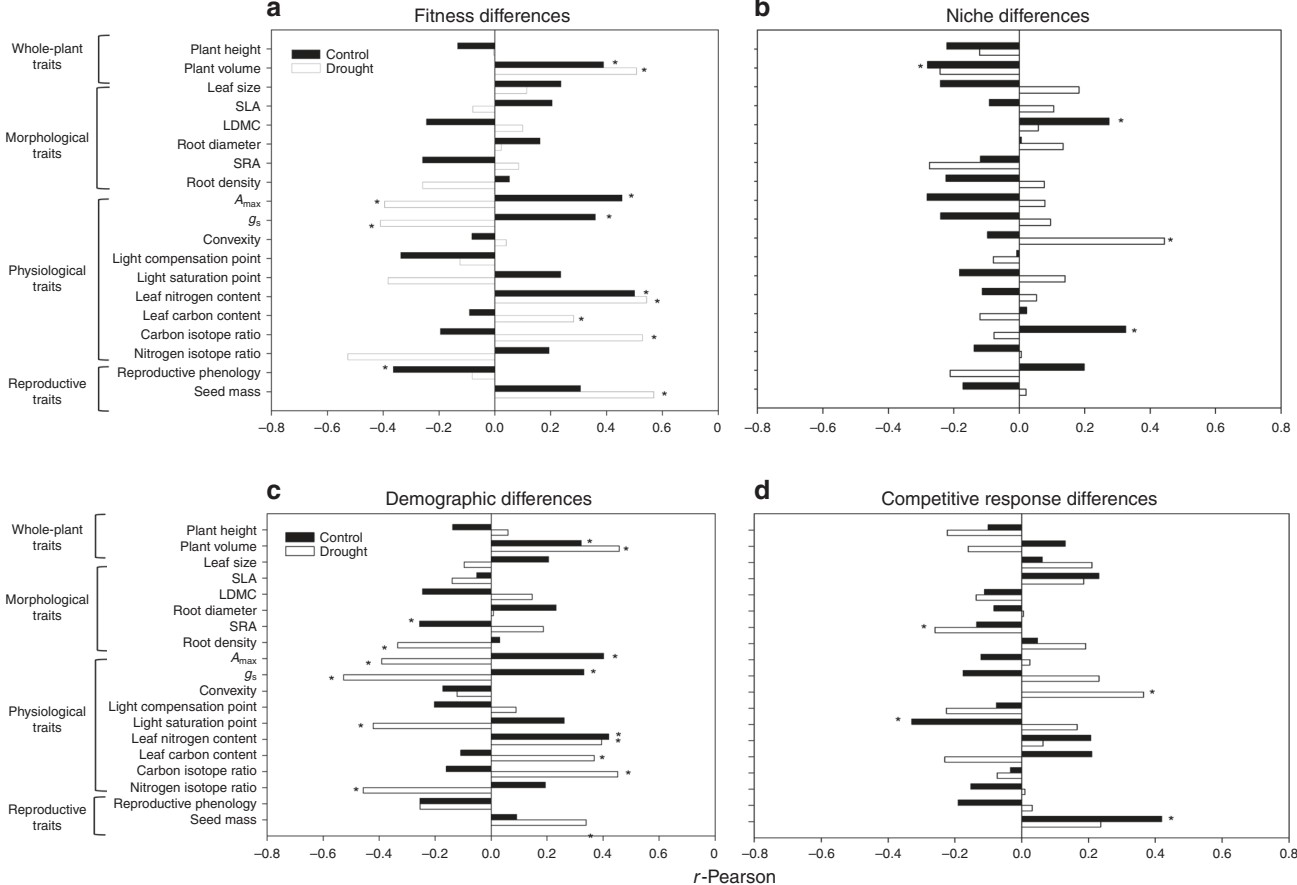

**Fig. 1** Functional traits as predictors of the determinants of competitive outcomes. Horizontal bars denote single $r$-Pearson values for correlations between the 19 functional traits considered in this study and average fitness (**a**) and stabilizing niche differences (**b**) as a function of climatic conditions (control and drought treatments with black and white bars, respectively). Trait correlations with the two components of fitness differences, the demographic ratio and the competitive response ratio, have been also represented (**c** and **d**). The functional traits that exercised a significant effect ($p < 0.05$) following Benjamini–Hochberg corrections for multiple comparisons have been marked with asterisks

was positively correlated with those trait values maximizing plant growth (i.e. higher values of $A_{max}$ and $g_s$). However, this wasteful resource-use typical of fast-growing species with acquisitive and exploitative strategies was negatively correlated with average fitness differences under drought conditions. Instead, competitive superiority was positively correlated with higher water use efficiency (i.e. less negative carbon isotopic discrimination— $\delta^{13}C$ values), more sclerophyllous leaves (i.e. with higher leaf C content) and bigger seeds (Fig. 1a). Further analyses showed that of the two components of the average fitness differences, functional traits were a better predictor of the demographic ratio

$[(\eta_j-1)/(\eta_i-1)]$ than the competitive response ratio $\left[\sqrt{\left(\frac{\alpha_{ij}\alpha_{ii}}{\alpha_{jj}\alpha_{ji}}\right)}\right]$

(a definition of these ratios is included in the section "Methods") for both climatic treatments (Fig. 1c, d).

The fact that drier conditions promote the competitive dominance of plants with a conservative resource-use strategy (i.e. with low SLA and/or high-density tissues) and a more efficient use of water is a result that is expected according to previous observational studies[25,26]. The competitive superiority of large-seeded species in drier years is often attributed to the recognized benefits of bigger seeds for a successful seedling establishment and growth under more stressful conditions[27–29]. Much less documented is the importance of functional trade-offs to determine species coexistence by partitioning environmental heterogeneity in temporal and spatial components. Similarly to previous studies[20,30], we found that a physiological trade-off between growth rate and low-resource tolerance determined competitive superiority under either moister or drier conditions. This might explain why species with higher photosynthetic rate and morphological leaf traits associated to a resource-acquisition strategy, such as *Calendula arvensis*, were more competitive under control climatic conditions but reduced strongly its average fitness in the drought treatment (Fig. 2 and Supplementary Table 1). The opposing case occurred for *Bromus madritensis* (with more sclerophyllous, small-sized leaves and low values of $A_{max}$ and $g_s$), which exhibited higher competitive superiority under more stressful conditions (Fig. 2 and Supplementary Table 1).

Critically, our study has detected the importance of single traits as drivers of stabilizing niche differences. Under average rainfall conditions, species stabilized their coexistence by presenting a

diversity of functional strategies related to resource partitioning (reflected here as differences in leaf dry matter content, LDMC) and water-use efficiency (i.e. differences in $\delta^{13}C$; Fig. 1b). Conversely, differences in plant volume reduced significantly niche differentiation, suggesting that the presence of a big species in the neighborhood does not allow another smaller species to exploit its own niche. Counter to expectations, none of the traits associated with a conservative water-use strategy were correlated with niche differences under the drought treatment. Instead, light curve convexity (a physiological trait representing the non-linear relationship between photosynthetic rate and light availability) was the only functional trait stabilizing species coexistence under stressful conditions (Fig. 1b). Light curve convexity is related to the ability of plant species to gain carbon under different irradiances[31], suggesting that how species partition light capture daily creates niche opportunities for coexistence not previously reported. Although the exact mechanism by which this relationship occurs is unclear, it is likely that time for capturing light is correlated with other set of physiological traits[32] more directly related to water and nutrient uptake. We definitely encourage further studies to delve into the processes by which this physiological behavior stabilizes population dynamics.

Our results also highlight the existence of complex interactions between climatic conditions and the influence of single functional traits on the determinants of competitive outcomes. Specifically, we found that water use efficiency (measured as $\delta^{13}C$) promoted stabilizing niche differences under control conditions but was a key component of competitive dominance (via demographic ratio differences) under drought (Fig. 1a, b). This result indicates that single traits might have opposing effect on the assembly of ecological communities depending on climatic conditions (see similar results for phenological differences in[14], but referred to a single environment). As a result of differences in this physiological trait, species in our experiment with the lowest water use efficiency (i.e. the most negative values of $\delta^{13}C$), such as *C. arvensis* and *Diplotaxis erucoides* decreased their competitive ranks under drier conditions (Fig. 2) because water limitation acted as an environmental filter reducing its fecundity and seed germination rates[2]. However, the inherent low water use efficiency of these species did not prevent them to coexist with other species under control conditions because allowed to have positive population growth rates at low relative abundances (i.e.

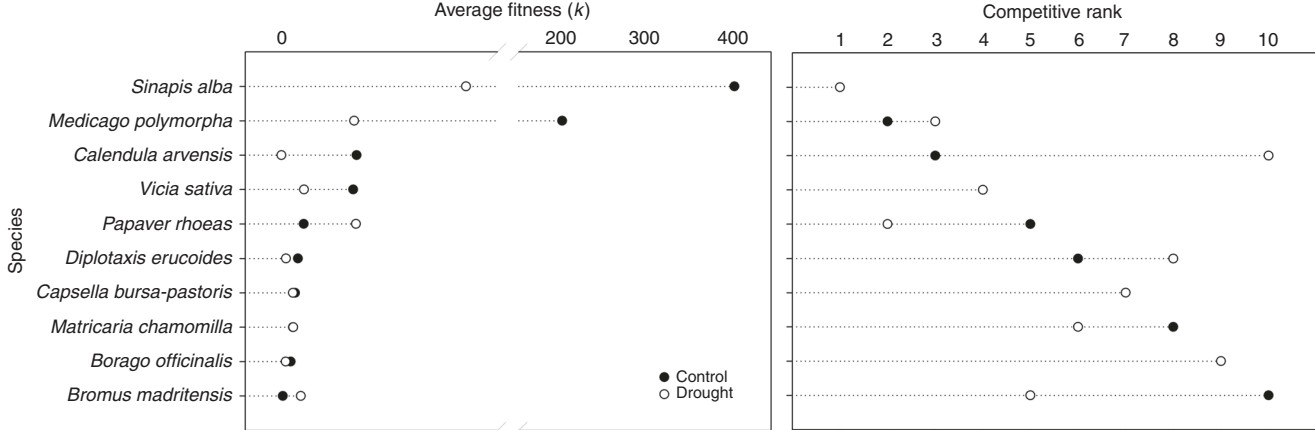

**Fig. 2** Estimates of species' average fitness and competitive ability for the two climatic conditions. Species' average fitness (left side) were obtained from experimentally derived species vital rates and competitive coefficients using Eq. (3).). Average fitness in the absence of niche differences served to rank species according to their competitive ability (right side). Black circles correspond to the control treatment and white circles to drought. The competitive rank panel serves to observe competitive trade-off between climatic treatments as species did not perform equally in both treatments. Some clear examples are *Calendula arvensis* and *Bromus madritensis*. In the competitive rank panel, only white circles are shown in those species that attained the same position in both climatic treatments

the population signature of stabilizing niche differences) (Supplementary Table 2).

Besides these univariate relationships, we also performed multi-trait models, which indicated that the best descriptors of both niche and fitness differences also include other phenotypic attributes that were not previously selected in the univariate trait models (e.g. plant height, some leaf, and root characteristics, and other light-curve parameters; Supplementary Table 3). The inclusion of these additional traits into the multivariate models reinforces the recent idea that species coexistence has a multi-dimensional nature[14]. Yet, the fact that multi-trait models did not select clear different suites of functional traits for each climatic treatment suggests that these multiple dimensions underlie mechanisms of diversity maintenance beyond the direct impact of rainfall modification. For instance, they could emerge by experiencing a similar suite of multitrophic interactions, such as leaf pathogens, herbivores, or mutualistic pollinators. It is also possible that all these traits selected by multi-trait models cover most of the dimensionality of competing species but they combine in a different way depending on the climatic conditions. The techniques used here do not allow unfortunately distinguishing between these different possibilities and, therefore, it would be desirable to develop novel techniques to address how different plant functional traits combine in non-additive ways to promote the determinants of competitive outcomes.

**Effects of drought-induced phenotypic plasticity on species coexistence.** Our experimental drought induced significant changes in almost all traits considered in this study (Supplementary Table 1). We firmly believe that the source of these changes came from phenotypic plasticity rather than from local adaptation, as we used the same well-mixed seed pool for both climatic treatments. The most obvious change of our experimental treatment was a delay in the peak of productivity between 18 and 28 days (Supplementary Table 1). Likely as a consequence of this imposed later phenology, nearly all species reduced their average plant height and many of them experienced a drastic decline in their occupation volumes (up to two orders of magnitude) under these more stressful conditions (Supplementary Table 1). This drought-induced reduction in plant performance was somehow expected since annual plants had less time for growing, and a shorter phenology commonly results in smaller individuals[14,24].

Moreover, drought significantly reduced average fitness for most species, with the exceptions of two species (*B. madritensis* and *Papaver rhoeas*), which increased their competitive abilities under these drier conditions (Fig. 2). These drought-induced shifts in species competitive abilities caused clear differences in the competitive hierarchy (measured as a species ranking in average fitness) when the two climatic treatments were compared (Fig. 2). Our results critically demonstrate that these changes can be attributed to interspecific differences in the degree of phenotypic plasticity. Spearman correlations for ranked data showed that those species with higher plasticity in one above-ground and one below-ground morphological trait (LDMC and root diameter, respectively) experienced less of an average fitness reduction across treatments and therefore maintained higher competition rankings (Table 2). In addition, generalized spearman rank correlations extended this relationship to plasticity in several morphological (leaf area), physiological ($A_{max}$, light curve convexity) and whole-plant traits (height) ($r_s^2 = 0.35$, $p < 0.001$; see Supplementary Table 4 for more details). Meanwhile, species exhibiting a relatively low phenotypic plasticity in these traits varied strongly their position in the competitive hierarchy depending on the treatment (being either good or bad

competitors). The positive effect of phenotypic plasticity on maintaining fitness homeostasis across variable environments[22] can be exemplified by the results obtained for *Sinapis alba*. This Brassicaceae species exhibited a combination of functional traits and phenotypic plasticity that enabled it to keep the first position in the competition ranking under the two climatic treatments (Fig. 2). Under control rainfall conditions, individuals of *S. alba* displayed a functional syndrome related with high carbon and nutrient acquisition, whereas those individuals subjected to drought drastically changed their physiology and produced smaller and denser leaves (i.e. with lower leaf area and higher dry matter content) and thicker roots (Supplementary Table 1). This large plasticity in multiple traits allowed *S. alba* to change from an acquisitive to a more conservative strategy, which might be interpreted as an adaptive mechanism for a more efficient use of resources and stronger tolerance to water limitation[33–35].

Interestingly, we also observed that the species' ability to exploit distinct niches in both climatic conditions was correlated with interspecific differences in phenotypic plasticity. Results from Mantel test showed that plasticity in light curve convexity and reproductive phenology promoted stabilizing niche differences, whereas plasticity in leaf nitrogen content and light saturation point produced the opposite pattern (i.e. they were correlated negatively with changes in niche differences; Table 2). Partial Mantel test did not select further traits at the multi-trait level. The positive relationship found between plasticity in light curve convexity and niche differences is congruent with what we have documented above for interspecific average differences (Fig. 1b), and the same occurs with plasticity in reproductive phenology compared to what was found in previous studies[14,24]. Our results underline therefore the increasing support of these two traits (light curve convexity and reproductive phenology) as stabilizing drivers of competitive dynamics between pairs of species, despite being of very distinct nature: one is a physiological trait acting at a daily temporal scale whereas the other is a reproductive trait that influences on the entire life cycle of the plant. What is surprising is to find that two physiological traits (leaf N and light saturation point) consistently correlated here and in another previous study[14] with average fitness differences between species pairs (Fig. 1a), are now acting as destabilizers of species coexistence. In practical terms, these distinct effects produce the same outcome, that is the dominance of the competitive superior species, but our study highlights the difficulty to infer the specific mechanism from just observing trait differences. What is clear regardless of these complex relationships, is that our study experimentally report the linkages between phenotypic plasticity and the determinants of competitive outcomes, supporting the theoretical expectation that phenotypic plasticity can both promote and hinder coexistence by modifying the amount of stabilizing niche and average fitness differences between species[19].

**Implications of competitive interactions for trait assembly patterns.** With niche and fitness differences being affected by multiple traits and their plasticity, a remaining question worth asking is how all this web of complex relationships translates into community trait patterns. In other words, do coexisting species and non-coexisting species differ in their trait differences? Whether coexisting pairs have greater or lesser trait differences than non-coexisting pairs can be used to test whether these phenotypic differences ultimately favor or impede coexistence[14]. To ask this question, we first compared the field-parameterized stabilizing niche differences $(1-\rho)$ to the average fitness differences $\left(\frac{\kappa_j}{\kappa_i}\right)$ for each species pair in the two treatments, which predict the outcome of competition at the neighborhood scale of

**Table 2 Effects of phenotypic plasticity on maintaining competitive dominance and varying niche differences**

| | Functional traits | Changes in competitive hierarchy (z-scores and p-values) | Changes in niche differences (r-values and p-values) |
|---|---|---|---|
| Whole-plant traits | Plant height | −0.581, 0.561 | −0.081, 0.645 |
| | Plant volume | −0.079, 0.937 | 0.057, 0.375 |
| Morphological traits | Leaf size | 0.785, 0.432 | 0.011, 0.454 |
| | Specific leaf area | 1.131, 0.258 | −0.038, 0.584 |
| | Leaf dry matter content | **−2.248, 0.025** | 0.066, 0.407 |
| | Root diameter | **−1.904, 0.048** | 0.061, 0.398 |
| | Specific root area | 0.003, 0.992 | −0.177, 0.788 |
| | Root density | −0.944, 0.345 | −0.211, 0.841 |
| Physiological leaf traits | Max. Photosynthesis | −0.143, 0.886 | −0.104, 0.754 |
| | Stomatal conductance | −0.514, 0.609 | 0.084, 0.381 |
| | Convexity | 0.631, 0.527 | **0.429, <0.001** |
| | Light compensation point | −1.369, 0.170 | 0.176, 0.192 |
| | Light saturation point | −1.391, 0.164 | **-0.403, 0.976** |
| | Leaf nitrogen content | −0.962, 0.335 | **-0.287, 0.955** |
| | Leaf carbon content | −1.073, 0.283 | 0.115, 0.233 |
| | Carbon isotope ratio | −0.047, 0.962 | 0.013, 0.439 |
| | Nitrogen isotope ratio | −0.874, 0.393 | 0.245, 0.078 |
| Reproductive traits | Reproductive phenology | −0.683, 0.539 | **0.368, <0.0001** |

The information gathered in this table are the results from the correlations between species' differences in trait plasticity induced by drought and: (i) the maintenance of competitive hierarchy (first column; Spearman correlations for ranked data); and (ii) changes in niche differences (second column; Mantel tests). Bold letters indicate those significant relationships at p < 0.05. Spearman correlations for ranked data involve a one-tailed analysis whereas Mantel tests involve a two-tailed analysis. Seed mass is not included as it did not present differences in species' average trait values between treatments

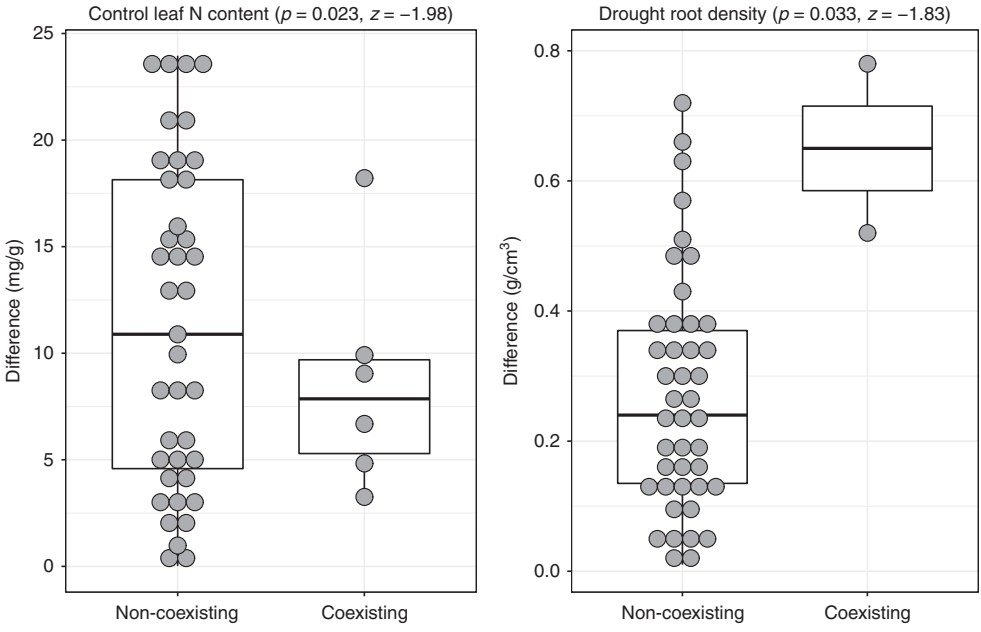

**Fig. 3** Significant average trait differences between predicted coexisting and non-coexisting species pairs. Pairs predicted to coexist represented by dots are more similar in Leaf N content under climatic control conditions but more dissimilar in root density under drought (Wilcoxon signed-rank test; p < 0.05). The other trait differences were not significant. Following boxplot representation standards boxes summarize median, first, and third quartile, while whiskers indicate value limits (minimum and maximum) for coexisting and non-coexisting pairs

species interactions. The condition for coexistence (i.e. when stabilizing niche differences overcome average fitness differences between competitors, $\rho < \frac{\kappa_i}{\kappa_j}$ being species j the superior competitor) was met for six pairs under control conditions and two species pairs under drought (Supplementary Table 2; more details in ref. [2]).

Wilcoxon signed-rank test showed that the six species pairs predicted to coexist under control rainfall conditions had significantly smaller differences in leaf N content than non-coexisting pairs (Fig. 3). This difference suggests that leaf N

content is more strongly related to average fitness differences than to stabilizing niche differences (Fig. 1). Therefore, since leaf nitrogen content is hindering coexistence, species must differ little in this trait in order to coexist. In fact, this similarity was true for the predicted coexistence of superior competitors (e.g. *S. alba* coexisting with *Medicago polymorpha*) as well as for the inferiors (e.g. *Capsella bursa-pastoris* coexisting with *Borago officinalis*). Wilcoxon signed-rank test also showed that the two species pairs predicted to coexist in the drought treatment (Supplementary Table 2) had significantly greater differences in root density than

non-coexisting pairs. Although the low sample size of coexisting pairs obligates to take this result with extreme caution, it is consistent with the multi-trait models showing that root density is more strongly related to stabilizing niche differences than to average fitness differences (Supplementary Table 3). Root density had hence an opposing effect on coexistence compared to leaf nitrogen content. This result implies that species with maximized differences in root density are more likely to coexist, supporting the relevant (often ignored) role of below-ground attributes as drivers of niche differentiation via soil resource partitioning (e.g. refs. [16,36]).

**Conclusions and climate change predictions**. By combining a field-competition experiment with demographic models of community assembly (following Chesson's framework[12,13]), we provide strong evidences that species coexistence is maintained by complex and interrelated processes involving multiple axes of trait variation associated with the three leading functional dimensions (i.e. plant economics, light interception, and reproductive ability[37–39]). In these complex relationships, we support the theoretical expectation that both average trait differences between species and their ability to modify their phenotypes in response to environmental heterogeneity (i.e. phenotypic plasticity) are equally important to determine the maintenance of species diversity across contrasted climatic conditions because both components of phenotypic variation are directly connected with the amount of stabilizing niche and average fitness differences between competitors. However, our study simultaneously highlights that most of the rich diversity of observed functional mechanisms governing community assembly processes hardly arise when evaluating just trait information from co-occurring or coexisting species. Therefore, it is paramount to rethink how information on functional traits needs to be used in order to properly summarize and predict the plethora of interactions that species establish with the environment and with other competitors.

Plant physiology is often overlooked in plant competition and coexistence studies (e.g. refs. [11,14]) given the difficulties to obtain measures of time-consuming traits across a broad range of species and environmental conditions (see ref. [20] for a rare exception). However, our results highlight that this common practice imposes a strong limitation because physiological traits accounts for key dimensions of plant population dynamics and coexistence that are not captured by morphological and reproductive traits. Critically, our study detected univariate relationships of trait differences with stabilizing niche differences thanks to the quantification of physiological traits related to water-use ($\delta^{13}C$) and light-use (light curve convexity) efficiency. These physiological traits promoted coexistence in our experiment under different climatic conditions by limiting species dominance when common and buffering them against extinction when rare. Moreover, other physiological traits related to carbon acquisition, such as maximum photosynthetic rates ($A_{max}$) and stomatal conductance ($g_s$) acted as different mechanisms of species coexistence creating trade-offs in species' fitness across climatic conditions, which might ultimately promote species turnover over space and time[20]. The importance of physiological traits make us aware that it is worth exploring in further studies other relevant traits not included here (such as those involved in plant interactions with other trophic levels including herbivores, pollinators and microorganisms[40]), as well as considering broader ranges of environmental or management conditions (such as differences in grazing intensity or in soil nutrient availability).

Finally, our experimental treatment obeys to the necessity to understand how the increased frequency of episodes of intense drought predicted by climate change models[23] will threaten plant diversity and will induce changes in species composition. Our findings suggest that drier conditions will reduce diversity[2] and will shift community dominance towards functional profiles of slow-growing species with more sclerophyllous leaves, bigger seeds and a more efficient use of water (Fig. 4). Conversely, high-growing species with opposing traits (i.e. with a more exploitative strategy) will be potentially excluded from the community under more stressful conditions unless they exhibit phenotypic plasticity in such way that allow them to maintain fitness homeostasis and to reduce niche overlap with competitors in response to drought (Fig. 4). These predictions come from a robust theoretical and modeling approach in which we have significantly advanced our fundamental understanding on how functional traits and their plasticity affect both the species' competitive ability and their ability to exploit distinct niches across contrasted environments.

## Methods

**Theoretical background for quantifying niche and fitness differences**. Our experiment was designed to parameterize a mathematical model that captures the demographic parameters of plants and their competitive interactions into direct and indirect components of population growth rates (i.e. niche and fitness differences[41,42]). This approach has been used in previous studies to accurately predict competitive outcomes between annual plant species[24]. Population growth is described as

$$\frac{N_{i,t+1}}{N_{i,t}} = (1 - g_i)s_i + \frac{\lambda_i g_i}{1 + \alpha_{ii} g_i N_{i,t} + \Sigma_{j=1}^{S} \alpha_{ij} g_j N_{j,t}} \tag{1}$$

where $\frac{N_{i,t+1}}{N_{i,t}}$ is the per capita population growth rate, and $N_{i,t}$ is the number of individuals (seeds) of species $i$ prior to germination in fall of year $t$. Changes in per capita growth rates depend on the sum of two terms. The first describes the proportion of non-germinated seeds $(1 - g_i)$ that survive in the soil seed bank ($s_i$). The second term describes how the per germinant fecundity in the absence of neighbors ($\lambda_i$) declines with the density of competing conspecific $\left(g_i N_{i,t}\right)$ and heterospecific $\left(\Sigma_{j=1}^{S} g_j N_{j,t}\right)$ neighbors. These neighbor densities are modified by the interaction coefficients that describe the per capita effect of species $j$ on species $i$ $\left(\alpha_{ij}\right)$, as well as of species $i$ on itself ($\alpha_{ii}$).

First, we followed the approach proposed by Chesson[13] to describe niche differences $(1 - \rho)$ for this model of population dynamics between competing species.

$$1 - \rho = 1 - \sqrt{\frac{\alpha_{ij} \alpha_{ji}}{\alpha_{jj} \alpha_{ii}}} \tag{2}$$

The stabilizing niche differences reflect the degree to which species limit individuals of their own species relative to heterospecific competitors. $1 - \rho$ is 1 when species have no interspecific effects (i.e. zero probability of niche overlap, favouring thus species coexistence), and it is 0 when species limit individuals of their own species and their competitors equally (i.e. maximum niche overlap; no possibilities of coexistence unless species are equivalent competitors).

Second, we calculated the average fitness differences between each pair of competitor species $\frac{\kappa_j}{\kappa_i}$ as follows:

$$\frac{\kappa_j}{\kappa_i} = \frac{\eta_j - 1}{\eta_i - 1} \sqrt{\frac{\alpha_{ij} \alpha_{ii}}{\alpha_{ji} \alpha_{jj}}}. \tag{3}$$

The species with higher value of $\kappa$ (either species $i$ or species $j$) is the competitive dominant, and it has the potentiality to exclude the other species in absence of niche differences. This equation shows that $\frac{\kappa_j}{\kappa_i}$ results from the interaction of two fitness components: the demographic ratio $\left(\frac{\eta_j - 1}{\eta_i - 1}\right)$ and the competitive response ratio $\left(\sqrt{\frac{\alpha_{ij} \alpha_{ii}}{\alpha_{ji} \alpha_{jj}}}\right)$. The former term describes the degree to which the species $j$ produces more seeds per seed loss due to death or germination than the species $i$. The second component describes the degree to which the species $i$ is more sensitive to both intra and interspecific competition than the species $j$. Note that the same interaction coefficients defining niche differences are also involved in describing the competitive response ratio, although their arrangement is different.

**Study area and experimental design**. We carried out a common-garden experiment at grasslands located in southwestern Spain (La Hampa station; 37° 16′58.8″N, 6°03′58.4″W; 72 m a.s.l.), over a whole growing season (from October 2014 to June 2015). Climate at the study area is Mediterranean-type

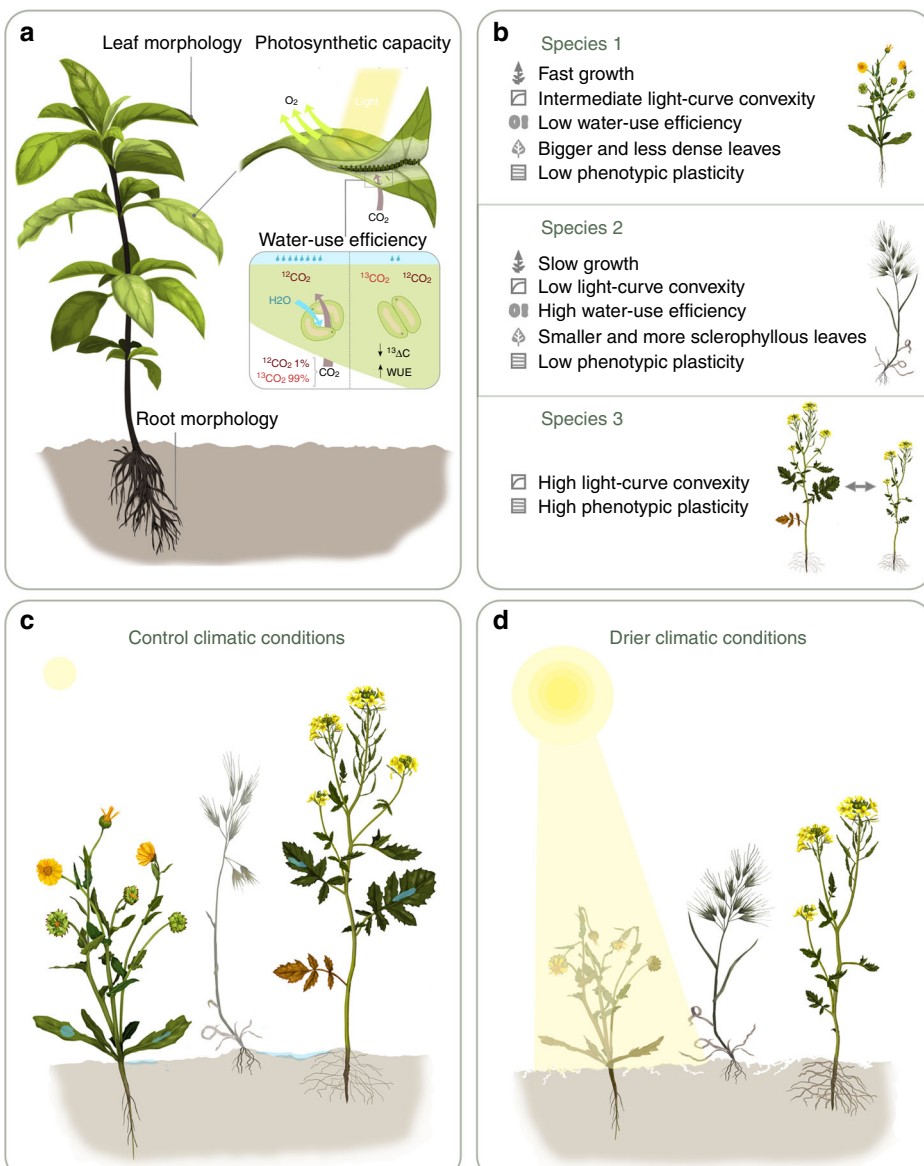

**Fig. 4** Schematic diagram representing the role of some relevant functional traits as drivers of species coexistence under contrasting climatic conditions. We represent functional attributes related to different plant dimensions (photosynthetic capacity, water-use efficiency, root and leaf morphology; panel **a**). Assuming a simple community composed of three plant species with different functional strategies (panel **b**), the outcome of plant competitive interactions strongly changes as a function of climatic conditions. Our results indicate that high-growing species with bigger and less dense leaves (e.g. species 1) will be more competitive under control climatic conditions (panel **c**). In contrast, low-growing species with opposing leaf trait values and a more efficient use of water (e.g. species 2) will be favored at times of low water availability (panel **d**), potentially excluding other species with contrasting functional attributes (e.g. species 1). Under these stressful conditions of aridity, competitive exclusion was avoided by the stabilizing effects of niche differences that arise from interspecific differences in light-curve convexity. Thus, species 2 and species 3 (with different values of light-curve convexity) coexisted at the drier scenario, likely due to their different ability to partition light capture daily. Finally, the species 3 also exhibited a high level of phenotypic plasticity in some morphological traits such as leaf dry matter content or root diameter, which conferred it a great competitive ability in both climatic scenarios and enabled it to share dominance with other good competitors in both moister (with species 1; **c**) and drier conditions (species 2; **d**). Paintings were created by "DharmaBeren Studio" (www.dharmaberen.com)

with cool wet winters and hot dry summers. Mean annual rainfall is 525 mm, with considerable inter-annual fluctuations, ranging from <250 mm in dry years to ≈900 mm in wet years. Mean annual temperature is 17.5 °C, with a mean monthly maximum of 25.3 °C (July) and a mean monthly minimum of 9.9 °C (January; mean values for the 2001–2018 period). We selected 10 common annual plants that naturally co-occur at the study site, spanning a wide spectrum of functional and phylogenetic diversity and including six of the most abundant families in Mediterranean grasslands (Supplementary Table 1). Seeds were provided by a commercial supplier (Semillas Silvestres S.L.) from populations located near the study site.

In September 2014, we established rectangular plots (0.65 × 0.5 m) separated by landscape fabric to control weeds within an 800 m$^2$ area, which had been

previously cleared of all vegetation and fenced to avoid mammal herbivory. To parameterize models of pairwise competition between the ten study species, we established a density gradient where each species competed against all others including themselves. To create this gradient, we randomly assigned each plot to a single species at four different densities (2, 4, 8 or 16 g/m$^2$ of viable seeds), using two replicates per density and species. Over this species background, we placed a grid (with four rows and five columns) that was used to sow seeds of the 10 species (with two replicates per each of them). In addition, to assess species' demographic performance without the influence of neighbors, we sowed 10 plots with the same grid of 10 species but without background neighbors.

Since seed germination and plant performance of annual species strongly depend on when major rains occur during autumn[2,43], we repeated the whole

experiment procedure at two distinct times in order to analyze the effect of contrasted climatic conditions on plant competitive dynamics, their phenotypic expression and their likelihood to coexist. Thus, we first sowed seeds in the ground just before the first major rains in early October 2014 (control treatment), and then we did the same during early December 2014 (drought treatment). This delay of 2 months (64 days) simulated a severe drought according to the climatic records available for the study area (daily data obtained from Seville airport meteorological station for the 1910–2010 period). Specifically, we documented a rainfall reduction of 206 mm in the drought treatment, representing a 38.7% of decrease in annual precipitation for that period 2014–2015. Overall, the total number of rectangular plots established for our experimental design was 180 (2 climatic treatments × 10 species × 4 background densities × 2 replicates = 160 plots, plus two climatic treatments × 10 plots without background competition = 20 plots).

**Quantification of species' vital rates and interaction coefficients.** First, we quantified the percentage of germination of viable seeds ($g_i$) by counting the number of seedlings emerged in at least four experimental plots per species (minimum one plot per density), and dividing this value by the total number of viable seeds sown. This fraction was further averaged across plots and included into the model as the species' germination term. Second, the specific' fecundities in the absence of neighbors ($\lambda_i$) and the per capita effect of each species on itself and their competitors (the interaction coefficients $\alpha$'s) were estimated by measuring seed production per plant ($F_i$) of all focal individuals in the 180 plots. For this purpose, we counted the total number of fruits produced per plant in the field, multiplied this value by the mean number of seeds per fruit (estimated in the laboratory in a subsample of fruits), and then corrected this number for seed viability. Third, the seed bank survival ($s_i$) was estimated by burying seeds on the surrounding area from October 2014 to August 2015 and determining their viability by the method described in ref. [24]. To quantify the competitive environment of each focal individual, we counted the number of competitor plants within a 7.5 cm radius just after germination[44]. At the end of the experiment, we used maximum-likelihood methods to fit both $\lambda_i$ and $\alpha_{ij}$ according to the following function:

$$F_i = \frac{\lambda_i}{1 + \sum_j \alpha_{ij} N_{j,t}} \qquad (4)$$

where $N_{j,t}$ was the number of germinated competitor individuals of the species $j$ surrounding focal individuals of the species $i$. For each target species $i$, we fitted a separate model jointly evaluating its response to individuals of all other species and itself. These responses were always positive meaning that species interactions were strictly competitive and, therefore, facilitation was not observed in our system. This approach fits a single per germinant fecundity in the absence of competition, $\lambda_i$, for each species $i$.

**Measurements of plant functional traits.** Plant functional measurements for each of the climatic treatment were taken on healthy adults of the 10 studied species. For that, we established 20 additional circular plots of 1 m² (10 per climatic treatment), where species competed in well-mixed communities at a total density of 8 g/m². Seeds in these plots were sowed at the same two times as the rest of the competition experiment and they were only used for destructive trait measurements, avoiding thus any interference of trait sampling with the quantification of species' vital rates and pairwise interaction coefficients. Within these circular plots, we measured 19 functional traits associated with plant physiology, morphology, phenology, and reproductive ability (Table 1). They were selected for covering the three leading dimensions of ecological variation among plants (i.e. plant economics, light interception, and reproductive ability[37–39], as well as for their recognized or assumed utility as response-traits to abiotic conditions[26,45] and their potential implication in plant competitive dynamics[11,14]. Plant sampling and trait measurements followed standard protocols recommended by[46].

At peak biomass, 30–50 individuals per species and treatment were randomly selected to measure two whole-plant traits (plant height and volume at the 95th quantile of their measured distribution), and three morphological above-ground traits (leaf size, specific leaf area, and LDMC; more details in Table 1). To estimate plant volume, we calculated its projected area from length and width measurements assuming an ellipsoid form, and calculated the volume of the spheroid. Leaf size, SLA, and LDMC were quantified following the protocol described by Gameir et al. [47]. Leaf size was quantified using an image analysis program (Image Pro-plus 4.5; Media Cybernetic Inc., Rockville, MD, USA).

Three morphological below-ground traits (root diameter, specific root area, and root density; Table 1) were measured in 3–7 individuals per species and treatment. In the field, we separated root systems from individuals of different species carefully. In the laboratory, we washed root samples to remove soil and then a representative sub-sample of fresh fine roots (<2 mm in diameter) was scanned at 1200 dpi. The digital images were used to determine length, area, and mean diameter of roots using specific image-analysis software (Winrhizo ver. 2003b, Regent Instruments Inc., Quebec, Canada). The root material harvested was immediately weighed, oven-dried at 60 °C for 48 h and then re-weighed.

Four chemical traits (leaf N and C content, as well as the bulk isotopic composition of these elements) were determined on three composed samples per species and treatment obtained by pooling the leaves previously used for above-ground traits, using a CHN elemental analyzer coupled to an isotope mass spectrometer at the laboratory facilities of Doñana Biological Station (EBD-CSIC).

Several physiological traits were measured as follows. Leaf-level photosynthesis ($A_{max}$) and stomatal conductance ($g_s$) were measured at the peak of vegetative growth in 6–9 individuals per species and treatment. Specifically, we measured leaf gas exchange with a portable LI-6400xt infrared gas-exchange analyzer system (Li-Cor Inc., Lincoln, NE, USA) at 400 ppm $CO_2$ concentration (using the Li 6400-01 $CO_2$ mixer), 1500 μmol m⁻² s⁻¹ light-saturating photosynthetic photon flux density (PPFD) and block temperature at 25 °C to match ambient air temperature. To determine projected leaf area of the measured leaves, they were further scanned and processed using the above-mentioned software, and this value was used to correct $A_{max}$. Moreover, three additional plants per species and treatment were used to construct light response curves following the procedure described in ref. [48]. For this purpose, photosynthetic gas exchange was measured at PPFD intensities of 1900, 1500, 1200, 900, 450, 200, 100, 50, 25, and 0 μmol m⁻² s⁻¹, using the same parameters indicated above for measuring $A_{max}$. We obtained three physiological traits from light response curves using Photosyn Assistant software 1.1.1 (Richard Parsons, Dundee, UK). This software models the photosynthetic response of leaves to variation in light level by a rectangular hyperbola following the quadratic equation presented by Chartier and Prioul[49]. The light compensation point is estimated from the intercept to $x$-axis and represents the light level at which leaf respiration and leaf photosynthesis result in a carbon balance equal to zero; the light saturation point is the light level at which the leaf reaches its maximal photosynthetic capacity; and, finally, the light convexity curve factor describes the progressive rate of bending between the linear gradient and the maximum photosynthetic value.

Finally, we measured two reproductive traits (reproductive phenology and seed mass). We used the date of peak fruiting as a measure of gross phenological differences between species following the procedure described by Kraft et al. [14]. We defined peak fruiting as the date when developing fruits outnumbered flowers on more than a 50% of the individuals of the species analyzed. Seed mass was quantified by weighing and averaging more than 1000 seeds per species. These weights were done when evaluating the amount of viable seeds needed to create the experimental density gradients. Seed mass is the only trait that did not differ between both climatic conditions. The 19 functional traits included in the study are listed in Table 1.

**Statistical analyses.** Before conducting any statistical analysis, an average of trait measurements across individuals per species and treatment was obtained, and species trait averages were log-transformed or square root-transformed as needed to fulfill assumptions of normality and homoscedasticity using Kolmogorov–Smirnov and Bartlett tests, respectively.

To evaluate the role of functional traits on the determinants of species coexistence under the two climatic treatments, we first used the field parameters of species' vital rates to calculate the stabilizing niche and average fitness differences between each pair of species (Eqs. (2) and (3)). Then, we tested for univariate correlations between these niche and fitness differences (and its two components, demographic, and competitive-response differences) and functional trait differences using Mantel tests, with the Benjamini and Hochberg correction for multiple comparisons. We also explored whether the two drivers of species coexistence were better described by multivariate sets of traits rather than single attributes. For this purpose, we conducted a model selection exercise in a Mantel framework by using the BEST routine in the PRIMER software package[50,51]. The BEST routine calculates Spearman's rho for all combinations of 1–19 functional trait differences and assesses the significance of the best-performing model using a permutation test. All these analyses were carried out using R (R Development Core Team, 2018). Additional code is not provided as all statistical analyses were done with common functions widely used in R.

To test the direct effects of how species were able to vary their phenotypic expression in response to the experimentally imposed drought (i.e. phenotypic plasticity), we conducted General Linear Models using the climatic treatment as the independent variable and each of the 19 functional traits considered in this study as dependent variables. These analyses were repeated separately for the two climatic treatments and for each of the 10 study species. The level of dependence among the 19 functional traits was determined using two complementary analyses: a principal component analysis (PCA) and Pearson's correlation analyses. Results from these analyses have been summarized and discussed in the Supporting information (Supplementary Tables 5 and 6, and Supplementary Fig. 1).

To evaluate the role of phenotypic plasticity on the variation of the determinants of species coexistence across both climatic treatments, we used two complementary approaches. For the relationships between trait plasticity and average fitness differences, we first ranked competitors in each climatic treatment according to their average fitness (Eq. (3)). With this information, we calculated changes in the position of the species along the competitive ranking between the two contrasted climatic conditions in absolute terms. Simultaneously, we computed species' plasticity to drought for each of the 19 traits as the difference in average trait values between the control and the drought treatment. We then correlated absolute changes in the competitive ranking to trait plasticity differences across species with the expectation to obtain a negative correlation (i.e. higher levels of plasticity favor fitness homeostasis across treatments and, therefore, the most

plastic species will not change their position along the competitive hierarchy in both climatic treatments[22]). The same analysis was conducted at the multi-trait level using generalized Spearman rank correlations.

For the relationships between trait plasticity and stabilizing niche differences, we do not have any previous expectation about how trait plasticity in response to drought affects changes in niche differences between pairs of species. Therefore, we correlated changes in niche differences between the control and the drought treatment to differences between species in trait plasticity in response to drought using Mantel tests, with the Benjamini and Hochberg correction for multiple comparisons. These are two-tailed tests, which allow evaluating whether drought-mediated trait plasticity increases or decreases niche differences between species pairs. Again, the same analysis was conducted at the multi-trait level using partial Mantel tests.

Finally, to evaluate the predicted outcome of competitive interactions between pairs of species in the experiment, we compared the magnitude of the estimated average fitness (Eq. (3)) and stabilizing niche differences between them (Eq. (2)). Stable coexistence at the scale of our experiment is predicted to occur when niche differences overcome fitness differences ($\rho < \frac{\kappa_j}{\kappa_i}$, where species $j$ is the fitness superior). Using this criterion, we used a series of Wilcoxon signed-rank tests to evaluate whether significant differences in each of the 19 evaluated traits emerge between coexisting pairs and non-coexisting pairs as consequence of direct competition. We also evaluated the role of indirect interactions, such as rock–paper–scissors dynamics on predicting the outcome of competitive interactions that occur when multispecies assemblages are considered. Specifically, we used a matrix inversion approach to algebraically solve for the unique equilibrium of species abundances for all combinations from 3 to 20 species (a total of 1012 assemblages). We then targeted which species assemblages were feasible and stable following the procedure described in ref. [2]. Unfortunately, we obtained a low replication of coexisting multispecies assemblages, which limited our ability to test the effect of interspecific trait differences on multispecies coexistence (Supplementary Table 2). Such a low degree of coexistence can be likely due to the fact that our experimental approach only allowed us to implement two contrasting levels of climatic conditions. Further studies considering a broader range of environmental conditions would be desirable to evaluate whether the large interannual climatic variability commonly detected in nature could increase the number of species that stably coexist. A summary scheme of all the statistical analyses conducted in this study is provided in Supplementary Fig. 2.

**Reporting summary**. Further information on research design is available in the Nature Research Reporting Summary linked to this article.

## Data availability

Estimation of species demographic parameters and pairwise competitive coefficients for both climatic treatments are available at Dryad Digital Repository https://doi.org/10.5061/dryad.5d1s9[52]. Average species means for the 19 functional traits included in this experiment are included in Supplementary Table 1.

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

## Acknowledgements

We thank to Yureli García de la Cruz, Juan S. Cara, Carmen Benítez, Elena Millán, and Jara Domínguez-Begines for their help during field sampling and plant measurements. We also thank to "DharmaBeren Studio" (www.dharmaberen.com) for creating the illustrations of Fig. 4. Funding support to conduct the experiment was provided by the Spanish Ecological Terrestrial Society (AEET, Jóvenes Investigadores grant 2014/2). I.M. P.-R. and L.M. were funded by a "Ramón & Cajal" contract (RYC-2013-13937) and an "Acción 6 UJA" fellowship (EI_RNM4_2017), respectively. O.G. acknowledges post-doctoral financial support provided by the European Union Horizon 2020 research and innovation program under the Marie Sklodowska-Curie grant agreement (No. 661118-BioFUNC). I.M.P.-R. and L.G.-A. also thank support from the MICINN projects DECAFUN (CGL2015-70123-R) and INTERCAPA (CGL-2014-56739-R).

## Author contributions

O.G. and L.G.-A. conceived and designed the study; O.G. and I.M.P.-R. conducted the statistical analyses; O.G., I.M.P.-R. and L.M. performed field sampling and plant measurements; I.M.P.-R. wrote the manuscript and O.G., L.M. and L.G.-A. contributed substantially to revisions.

## Additional information

**Competing interests:** The authors declare no competing interests.

