## [Peer Review File · Nature Communications]

Reviewers' comments:

Reviewer #1 (Remarks to the Author):

First of all, I like this piece. I went through the text a couple of times, and found the presentation is really clear to me. Linking functional traits and components of modern coexistence theory has been a hot topic. Several studies reported the differences in traits usually represented the fitness differences between species, which sometimes is counterintuitive. This work demonstrated why this happened. In addition, intraspecific trait variability has been considered critical for species coexistence as well, but the empirical work is less. Especially, phenotypic plasticity is strongly dependent on the environment organisms experience. Then the issue arises that how environmental conditions influence coexistence for the same set of species. Through putting all the above points together, using mature experimental design and theoretical derivation for annual plants (Levine's legacy), authors conducted their work under contrasting climatic conditions (drought). Various types of plant functional traits were measured: morphological, physiological and reproductive. This is an amazing dataset. They found that traits related to water- and light-use-efficiency and their plasticity promoted species coexistence across climatic conditions, which enhanced stabilizing niche differences and generated competitive trade-offs between species.

For myself, I have no problem, actually. However, considering the broad audience of the Journal, i.e. for readers who are not very familiar with the coexistence theory, I am wondering if authors could present a workflow chart of the analyses. In the text, authors used various approaches for different objectives, which could bring the chaos to readers. I would like to see such a workflow to show this information, and followed by the major concern solved. Anyway, good job.

Reviewer #2 (Remarks to the Author):

The manuscript by Perez-Ramos and colleagues uses an annual plant system to assess the value of traits and plasticity to understand fitness and niche differences. Overall, this is a very nice paper, that combines a simple coexistence model with an experimental assessment in two water treatments. The results are novel and interesting, and well worth publishing. I have a couple of thoughts that might benefit from consideration in the manuscript.

1) The concept of niche differences in this paper is really a local scale phenomenon. How do the results translate into spatial or temporal coexistence mechanisms?

2) This is an arid system and I wonder if there is facilitation occurring, especially in the lower density treatments. Can this be detected, or perhaps comments on?

3) In other papers using similar experimental annual plant systems there's actually been very little evidence of stabilizing coexistence (only a few species pairs show this), and in the current manuscript, it is difficult to determine how much coexistence is predicted. And if low, this deserves some discussion, maybe couple with point #1 above.

I did find some weakness in the writing and have included an annotated copy.

I hope that these comments help.

Point by Point letter detailing the responses to referees' comments of the manuscript entitled "Functional traits and phenotypic plasticity modulates species coexistence across contrasting climatic conditions" with reference NCOMMS-19-03775A

Reviewer #1

COMMENT 1: First of all, I like this piece. I went through the text a couple of times, and found the presentation is really clear to me. Linking functional traits and components of modern coexistence theory has been a hot topic. Several studies reported the differences in traits usually represented the fitness differences between species, which sometimes is counterintuitive. This work demonstrated why this happened. In addition, intraspecific trait variability has been considered critical for species coexistence as well, but the empirical work is less. Especially, phenotypic plasticity is strongly dependent on the environment organisms experience. Then the issue arises that how environmental conditions influence coexistence for the same set of species. Through putting all the above points together, using mature experimental design and theoretical derivation for annual plants (Levine's legacy), authors conducted their work under contrasting climatic conditions (drought). Various types of plant functional traits were measured: morphological, physiological and reproductive. This is an amazing dataset. They found that traits related to water- and light-use-efficiency and their plasticity promoted species coexistence across climatic conditions, which enhanced stabilizing niche differences and generated competitive trade-offs between species.

RESPONSE: We thank the reviewer for these positive comments. We also think this is a unique database. Indeed, it took great effort to put it together. Following *Nature communications* requirements, the whole data set related to estimations of species vital rates and species interactions coefficients is already included in the "Dryad Digital Repository", and information on average species means for the 19 traits measured in this study has been also included as a supplementary file (Supplementary Table 1). This is now explicitly indicated in the new version of the manuscript (lines 591-595), where we have inserted a "Data availability" section.

COMMENT 2: For myself, I have no problem, actually. However, considering the broad audience of the Journal, i.e. for readers who are not very familiar with the coexistence theory, I am wondering if authors could present a workflow chart of the analyses. In the text, authors used various approaches for different objectives, which could bring the chaos to readers. I would like to see such a workflow to show this information, and followed by the major concern solved. Anyway, good job.

RESPONSE: This is a very good suggestion. We have now provided a scheme detailing the workflow and the specific statistical analyses used for each of the objectives of our study. This scheme has been included in the new version of the manuscript as the supplementary Fig. 2. Anyway, we would have no objection in including this information into the main text in the case that the Editor considers it necessary.

Reviewer #2

COMMENT 1: The manuscript by Perez-Ramos and colleagues uses an annual plant system to assess the value of traits and placidity to understand fitness and niche differences. Overall, this is a very nice paper, that combines a simple coexistence model with an experimental assessment in two water treatments. The results are novel and interesting, and well worth publishing. I have a couple of thoughts that might benefit from consideration in the manuscript.

RESPONSE: We are very grateful for the highly positive comments provided by the reviewer. The specific changes incorporated in this new version of our manuscript following the reviewer's recommendations are discussed in detail below..

COMMENT 2: The concept of niche differences in this paper is really a local scale phenomenon. How do the results translate into spatial or temporal coexistence mechanisms?

RESPONSE: This is a very good comment. The concept of niche differences in this paper is the same than that referred in all previous papers using recent advances of coexistence theory (following Chesson's framework). As pointed out by the reviewer, this is commonly treated as a local-scale phenomenon, basically because this is the scale where species interactions occur. Thus, niche differences are estimated as the ratio of intraspecific interactions compared to interspecific interactions, reflecting the degree to which species limit individuals of their own species relative to heterospecific competitors (see equation 2 in the main text). Nevertheless, although stabilizing niche differences are referred at a local scale, they could potentially maintain diversity over larger scales if these were spatially or temporally homogeneous. Otherwise, the superior competitor in the absence of niche differences would exclude the rest of the community as predicted by using the Tilman's rule of the R^* (Tilman 1982)

Tilman, David (1982). *Resource competition and community structure*. Princeton: Princeton University Press

In the case that the environment is heterogeneous, as simulated in our study thanks to the implementation of two experimental climatic treatments, we have shown the existence of functional trade-offs between species that allow some species to dominate in one particular environment but not in another. This is a good example of a mechanism of species coexistence over broader temporal or spatial scales by promoting competitive dominance at one particular environment but not at the other.

In addition, if species would present niche differentiation in each particular site, that would in turn increase the diversity maintained at each location due to the coexistence of several competitors within each locality. This is again what happens in our experiment, where several species pairs are predicted to coexist under control climatic conditions but these pairs differ from those predicted to coexist under drought.

All these issues were already discussed in detail in the first version of our manuscript.

COMMENT 3: This is an arid system and I wonder if there is facilitation occurring, especially in the lower density treatments. Can this be detected, or perhaps comments on?

RESPONSE: This is again a good comment and we thank the reviewer for pointing it out. Our ecosystem is not an arid system but rather a Mediterranean prairie in which study species are well adapted to unpredictable inter-annual variation in rainfall amount. All estimations of interaction coefficients in our study were positive, meaning (according to equation 4) that all species reduced the seed production of focal individuals in the presence of neighbors. In other words, they were strictly competitive interactions, and we did not observe facilitation in our study system. It is true, however, that some competitive interactions were stronger than others (i.e. higher positive values). It is also true that some of them were closer to zero, meaning that the competitive effect of one species on another was close to be neutral but never facilitative. In order to clarify this issue, we have now included a sentence indicating the lack of facilitative interactions between the study species (lines 453-455). The matrix of intra and interspecific coefficients is available at Dryad Digital Repository: <https://doi.org/10.5061/dryad.5d1s9>

COMMENT 4: In other papers using similar experimental annual plant systems there's actually been very little evidence of stabilizing coexistence (only a few species pairs show this), and in the current manuscript, it is difficult to determine how much coexistence is predicted. And if low, this deserves some discussion, maybe couple with point #1 above.

RESPONSE: We have included in lines 584-589 a point of discussion about the degree of species coexistence predicted in our experiment. The number of species predicted to coexist is relatively low compared to the total species pool because we use a very strict metric of coexistence. Namely, we evaluated the number of species combinations that stably coexist at equilibrium (i.e. when time tends to infinity) under a homogeneous environment, either under control conditions or under drought. Unfortunately, our predictions did not take into account all the inter-annual variability that occur in our system, that seems to be one of the main putative mechanisms of diversity maintenance in our system. As now recognized in the new version of the manuscript, we would be able likely to obtain more predicted coexistence than that discussed in our original submission (lines 341-345) in the case we would have used a broader range of climatic variability. This potentially higher degree of coexistence would be a combination of niche differences and competitive trade-offs between species.

COMMENT 5: I did find some weakness in the writing and have included an annotated copy. I hope that these comments help.

RESPONSE: We would like to thank the reviewer for taking the time and the effort to go through the manuscript in careful detail. We have included all comments and changes in the revised version of the manuscript, which have been denoted by red color.

REVIEWERS' COMMENTS:

Reviewer #2 (Remarks to the Author):

Thank you for addressing my concerns. While I might have liked a little more thought about the limitations of the experiment (e.g., small scale, water limitation, etc.), I think it is a nice overall test and more than sufficient to influence our understanding of coexistence in the literature.